# 🤖 Open CaptchaWorld: A Comprehensive Web-based Platform for Testing and Benchmarking Multimodal LLM Agents

**Yaxin Luo**[*], **Zhaoyi Li**[*], **Jiacheng Liu, Jiacheng Cui, Xiaohan Zhao, Zhiqiang Shen**[†]

[1]VILA Lab, MBZUAI     [2]MetaAgentX

[*]Equal Contribution     [†]Corresponding Author

**Code & Data:** Open CaptchaWorld

## Abstract

CAPTCHAs have been a critical bottleneck for deploying web agents in real-world applications, often blocking them from completing end-to-end automation tasks. While modern multimodal LLM agents have demonstrated impressive performance in static perception tasks, their ability to handle interactive, multi-step reasoning challenges like CAPTCHAs is largely untested. To address this gap, we introduce **Open CaptchaWorld** 🤖 , the first web-based benchmark and platform specifically designed to evaluate the visual reasoning and interaction capabilities of MLLM-powered agents through diverse and dynamic CAPTCHA puzzles. Our benchmark spans 20 modern CAPTCHA types, totaling 225 CAPTCHAs, annotated with a new metric we propose: *CAPTCHA Reasoning Depth*, which quantifies the number of cognitive and motor steps required to solve each puzzle. Experimental results show that humans consistently achieve near-perfect scores, state-of-the-art MLLM agents struggle significantly, with success rates at most **40.0%** by Browser-Use Openai-o3, far below human-level performance (**93.3%**). This highlights Open CaptchaWorld as a vital benchmark for diagnosing the limits of current multimodal agents and guiding the development of more robust multimodal reasoning systems.

## 1 Introduction

Multimodal agents powered by large language models (LLMs) [42, 13, 27, 6, 5, 29, 9] are rapidly advancing toward real-world deployment, with the promise of automating tasks such as form filling, navigation, shopping and other interactions on websites. However, one major roadblock remains: CAPTCHAs. These human verification puzzles, designed to prevent bots from abusing web services, frequently prevent agents from completing real tasks, especially on high-value sites like e-commerce platforms or login pages. For agent-based systems to be truly deployable in the wild, solving CAPTCHAs autonomously must become a core capability.

Recent Multimodal LLMs (MLLMs) such as Openai-o3 [29], Claude3.7-Sonnet [4], and Gemini2.5-Pro [9] have demonstrated strong capabilities across a range of visual-language tasks, including object grounding [33, 48, 40], VQA [11, 14, 24, 37], and document analysis [25, 15, 52]. They can observe screenshots, interpret UI elements, and issue text or click-based commands. Yet these models are usually tested in static, one-shot benchmarks, lacking the multi-step, tool-using, and interaction-heavy dynamics found in CAPTCHA tasks. As a result, we still lack a reliable assessment of whether these models can reason and act like humans in complex, vision-guided interactions.

39th Conference on Neural Information Processing Systems (NeurIPS 2025) Track on Datasets and Benchmarks.

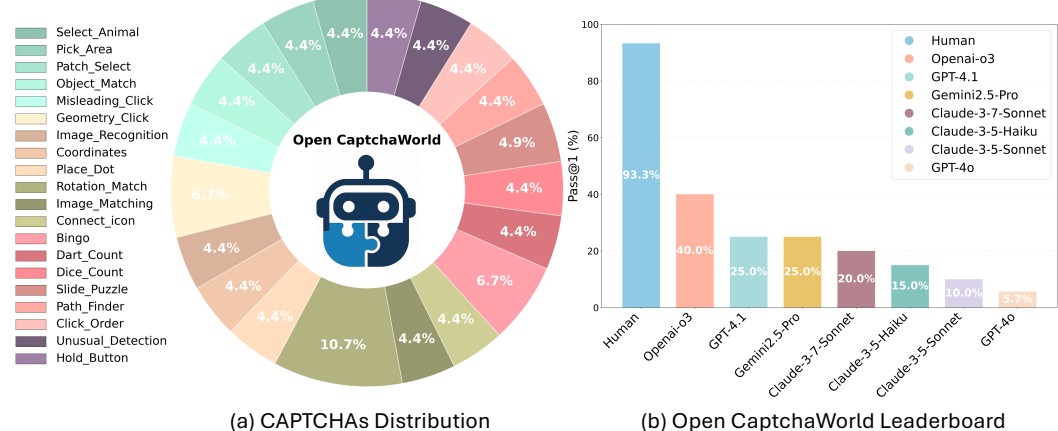

(a) CAPTCHAs Distribution  (b) Open CaptchaWorld Leaderboard

Figure 1: **Open CaptchaWorld data distribution and MLLMs performance plot.**

Despite the explosion of agent benchmarks, most systematically filter out CAPTCHAs. VisualWebArena [17] and AgentBench [21] simulate realistic environments but discard pages with CAPTCHAs [45]. Traditional CAPTCHA-solving work (e.g., Deep-CAPTCHA [28], Breaking reCAPTCHAv2 [32]) treats them as static perception tasks solvable by CNNs or object detectors, ignoring the sequential planning and interface state dynamics. This leaves a crucial evaluation gap: no benchmark tests whether MLLM agents can handle CAPTCHAs in a closed-loop, interactive setting that mimics real-world browsing.

To close this gap, we introduce **Open CaptchaWorld**, a web-based benchmark designed to assess whether agents can autonomously solve modern CAPTCHAs through perception, reasoning, and multi-step interaction. Our benchmark includes drag-based, sequence-click, slider alignment, and counting-based puzzles, all designed to be intuitive for humans but challenging for current agents. Unlike prior work that filters CAPTCHAs out, we embrace them as essential obstacles for agent robustness and autonomy. Our benchmark consists of 20 diverse CAPTCHA types, the amount of each types will be continuously increasing and a novel metric called **CAPTCHA Reasoning Depth**, which quantifies how many cognitive and motor steps are needed to solve the task. Despite its modest size, Open CaptchaWorld represents a highly challenging and realistic benchmark for agent-based multimodal reasoning, owing to its interactive nature, step-by-step decision requirements, and high variance in visual-cognitive complexity. All puzzles are tested in a real browser loop, where agents must perceive screenshots and issue clicks or key actions until the task is complete. We evaluate a broad spectrum of the most advanced MLLM models equipped with browser-use tools [26], including Openai-o3, Claude3.7-Sonnet, Gemini2.5-Pro, and GPT-4.1 etc, find that success rates vary widely by puzzle type and depth. Notably, even top-performing agents lag behind humans by **-53.3%**.

Moreover, the benchmark is explicitly designed to test generalization and reasoning depth, not memorization from massive data. As our evaluations show, state-of-the-art agents perform far below human levels. **Our main contributions are as follows:** (1) We propose **Open CaptchaWorld**, the first open-source, large-scale, and long-term maintaining CAPTCHA benchmark for evaluating interactive multimodal agents using MLLMs. (2) We introduce **CAPTCHA Reasoning Depth**, a task-agnostic complexity measure capturing the multi-step reasoning burden of visual interaction puzzles. (3) We build a real web-based testing platform[1] and systematically evaluate state-of-the-art models in zero-shot settings, revealing large performance gaps compared to humans. (4) We provide insights into agent failure cases such as overthinking, over-segmentation and interface misunderstanding.

## 2 Related Work

The evolution of multimodal LLMs (MLLMs) such as Openai-o3 [29], Gemini2.5-Pro [9], and DeepSeek-V3 [43] has been driven by increasingly diverse benchmarks [1, 18, 20, 54, 6, 5], ranging from math [23], visual QA [12, 14, 24], to OCR-based reasoning [37]. To assess these models comprehensively, benchmarks like MMBench [22], MME [8], MMMU [50], and MM-Vet [49]

---

[1]https://huggingface.co/spaces/OpenCaptchaWorld/platform.

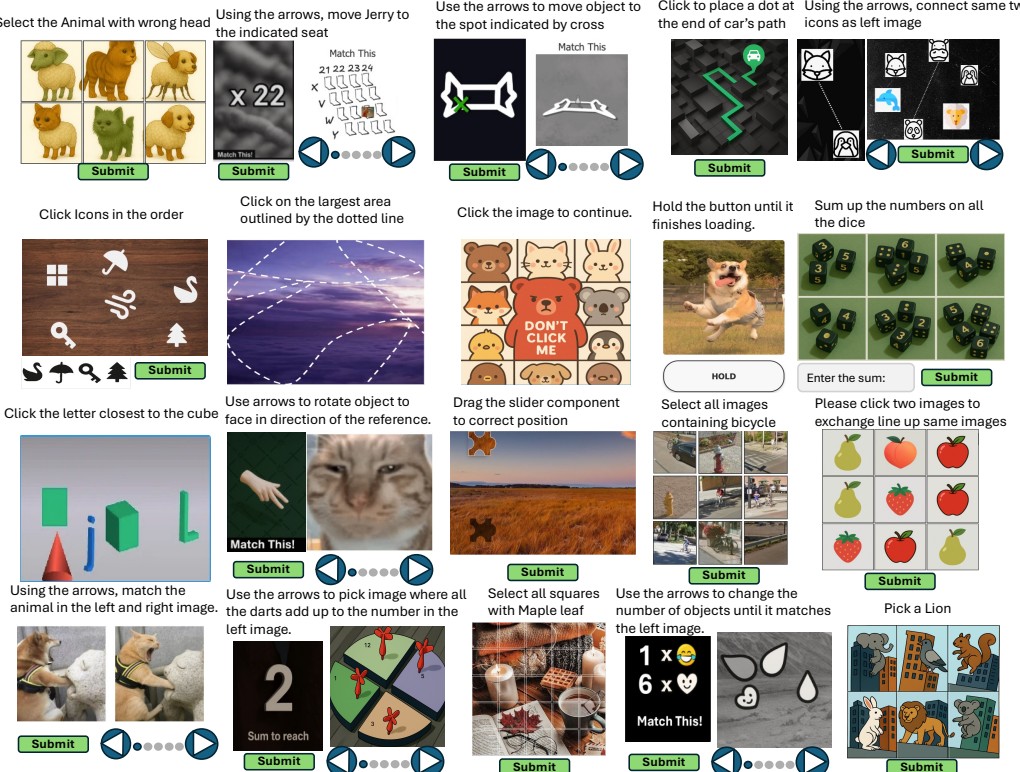

Figure 2: **Examples from Open CaptchaWorld.**

evaluate a wide range of MLLM capabilities. However, most assume a static, single-turn setup [47], limiting their ability to test dynamic, real-world interaction. To overcome this, recent work has explored LLM and MLLM agents operating in interactive environments [31, 39, 35], often with external tool use [51, 7, 10, 19, 34] and multi-step decision-making [46, 41, 36]. Benchmarks like SWE-bench [16] test an agent's ability to debug and patch codebases, while WebArena [53] and its multimodal extension VisualWebArena [17] require agents to interpret text and images to complete web-based goals. AgentBench [21] aggregates tasks across diverse domains, and ToolBench [17] isolates tool-use challenges. However, CAPTCHAs remain underexplored in this agentic paradigm. Existing solutions [28, 32] treat CAPTCHA solving as static vision tasks, ignoring interactive challenges like UI state tracking, fine-grained control, and sequential decision-making. In contrast, modern LLM agents integrate perception, reasoning, and action [46, 36], making them suitable for solving complex CAPTCHA puzzles in dynamic environments. Despite progress in multi-turn reasoning benchmarks, no open-source efforts target CAPTCHA solving in the way AgentBench [21] or VisualWebArena [17] test broader interactions. Our work fills this gap by introducing a web-based CAPTCHA benchmark where MLLM agents must perceive, plan, and act over multiple steps, providing a realistic testbed for evaluating agent robustness beyond static classification.

## 3 Open CaptchaWorld

Open CaptchaWorld is a carefully curated benchmark designed to evaluate multi-step, interactive visual reasoning CAPTCHAs that are hard for models but easy for humans to solve. Inspired by commercial CAPTCHA systems like Google's reCAPTCHA, Arkose Labs' Arkose MatchKey. We systematically design and annotate images to construct Open CaptchaWorld web-based benchmark for Multimodal Agents. All images are either drawn by human designers or generated by GPT-4o [30].

### 3.1 Open CaptchaWorld serves as a complement to Web Agent's benchmarks

With progress of Agent's development, the web agents will finally be deployed in real-world applications to automatically finish tasks on websites. However, we notice that previous research

usually ignores websites that contain CAPTCHAs, because tasks involving websites with CAPTCHA prevent agents from completing the task. However, those websites are usually more commercial and popular websites, which contain more real-life, day-to-day tasks. Besides web Agents, the existing benchmarks usually discard web pages that contain a CAPTCHA system when they construct their benchmarks [44]. However, in order to deploy web agents in the real world, the captcha cannot be easily ignored and skipped; we need to develop solutions for web agents to tackle this challenge.

To address this overlooked crucial challenge, Open CaptchaWorld is introduced as a dedicated benchmark explicitly targets web environments containing CAPTCHAs. Unlike prior datasets that filter out these interaction barriers [44], Open CaptchaWorld embraces them as necessary components for evaluating the readiness of web agents in real-world deployments. CAPTCHAs are not edge cases, which are commonly encountered in high-value, security-sensitive websites such as ticketing platforms, e-commerce portals, and account login flows. Bypassing them in evaluation leads to a misleading sense of agent competence. We systematically curate a diverse set of CAPTCHAs, spanning image-based selection, drag-and-drop mechanics and jigsaw alignment, etc. These scenarios go beyond static perception, which requires agents to combine multimodal understanding, memory across steps, and dynamic interaction with on-page elements. As such, this benchmark shifts the focus from single-turn prediction to interactive problem-solving, a key trait for practical usage.

## 3.2 CAPTCHA Reasoning Depth

To better characterize cognitive difficulty of puzzles in *Open CaptchaWorld*, we introduce a new metric called "*CAPTCHA Reasoning Depth*", which quantifies the number of reasoning and interaction steps a human must perform to solve a given CAPTCHA. Unlike traditional classifications that group puzzles by type (e.g., image selection, jigsaw, or drag tasks), reasoning depth offers a task-agnostic measure of complexity that aligns more closely with the multi-step nature of agent reasoning. We define CAPTCHA Reasoning Depth as the minimal number of atomic reasoning or decision-making steps required by a human or a model to arrive at a correct solution, where each step involves interpreting visual content, planning a subgoal, or executing a discrete interaction (e.g., a drag, click, or alignment operation). Formally, let a CAPTCHA be defined as a task $T$ requiring a sequence of operations. We define the CAPTCHA Reasoning Depth $D(T)$ as:

$$D(T) = \sum_{i=1}^{N} \mathbb{I}[s_i \in \mathcal{S}_T] \tag{1}$$

where $\mathcal{S}_T$ is the set of atomic steps needed to solve $T$, $s_i$ is an atomic reasoning or interaction step from a predefined checklist $\mathcal{C}$ (see Table 4), and $\mathbb{I}[\cdot]$ is the indicator function. Each $s_i$ contributes 1 unit of depth if the step is observed during the solution process. The checklist $\mathcal{C}$ includes categories such as visual perception, cognitive planning, motor control, and state monitoring.

For instance, a puzzle that asks user to "click on the fox" typically requires two steps: first, identify target object among distractors, and then perform click. In contrast, a drag-based jigsaw CAPTCHA may require identifying multiple part alignments, sequencing them appropriately, and dragging each piece to its correct location, leading to a reasoning depth depending on puzzle layout and ambiguity. To measure reasoning depth across the benchmark, we conducted a human annotation study where participants solved sample puzzles while verbally decomposing their thought process into atomic reasoning steps, guided by a set of heuristic rules (Table 4). We averaged the number of steps across annotators to estimate the reasoning depth per puzzle and measured inter-annotator agreement to ensure consistency. To compare with LLM agents, we also prompted Openai-o3 [29] and Gemini2.5-Pro [9] using the same heuristics (Fig 10). Fig 3 shows the human-estimated depth distribution, revealing broad cognitive diversity: an average depth of 2.94 with a standard deviation of 0.92. Each CAPTCHA type includes at least 10 diverse variants, varying in layout, icons, or interaction mode.

**Different Reasoning Depth Estimate Behavior Between Human and Models.** To better understand why MLLM models and humans provide different reasoning depth estimations shown in Fig. 3, we compare their thinking processes when analyzing the same CAPTCHA. Fig. 4 illustrates an example to this difference. For example, in a sequence-matching CAPTCHA, the human annotator simply identifies the icon order from reference image, searches for them in main panel, clicks each in sequence, and submits the answer, resulting in a depth score of 3. Humans focuses only on key goal-directed actions, compressing low-level perception and memory usage into intuitive, seamless behavior. In contrast, the Openai-o3 model oversegments the process. It lists granular

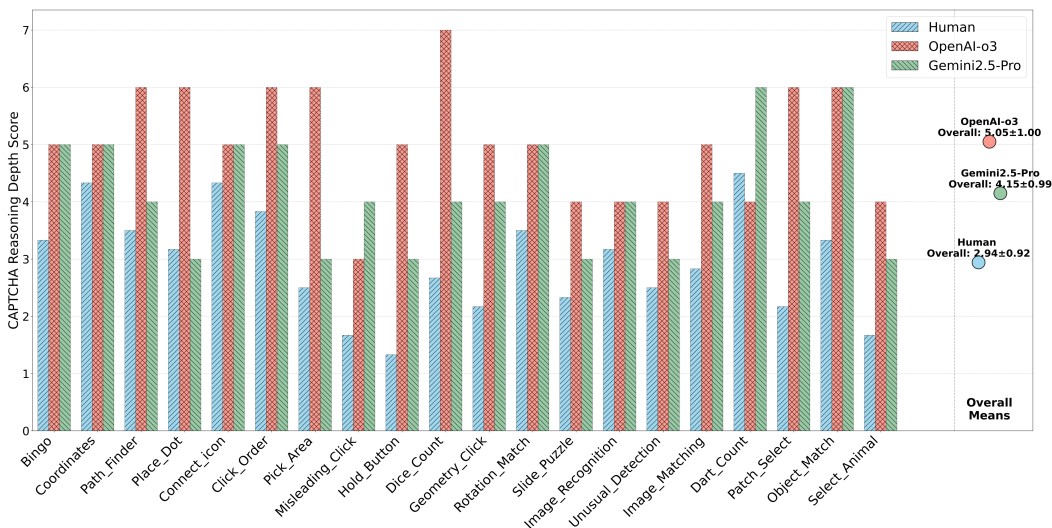

Figure 3: CAPTCHA Reasoning Depth Estimation by Human Annotators and Most Advanced Reasoning Models.

**Human Response**: Okay, the reference order image is at the bottom left and the order is duck, umbrella, key then tree. I will search those components in the main image, and then click these icons in previous order. Finally, submit my answer. Hence, the reasoning depth for me is 3.

**Click Icons in the order**

**O3 response**: The user has given a puzzle image along with a checklist. I need to break down the reasoning depth to ensure clarity in execution. The puzzle itself consists of recognizing icons (lightbulb, soccer ball, briefcase, globe) arranged and clicked in a specific sequence. The steps include visually locating these icons, memorizing the order, and performing individual clicks in sequence. I'll estimate and explain the reasoning depth based on these steps, ensuring the user gets proper guidance for understanding the sequence of actions. Solving this CAPTCHA requires first visually recognising the four distinct icons (light-bulb, ball, brief-case, globe), then memorising the reference sequence so you can reproduce it without relying on the guide. With that cue held in working memory, you plan the exact order in which to click the scattered icons, execute those clicks one by one, and after each press quickly check the interface feedback (e.g., highlighted icons and click counter) to confirm progress before moving on. Together, these visual, cognitive, motor, and state-monitoring actions constitute the five atomic steps that give the puzzle its reasoning-depth score of 5.

Figure 4: Thinking Process Comparison When Estimating CAPTCHA Reasoning Depth between human and Openai-o3 model.

steps such as recognizing each icon, memorizing their order, executing each click separately, and monitoring interface feedback after every action. This leads the model to assign a higher reasoning depth. The model treats each sub-action (e.g., "confirm progress" or "hold cue in memory") as a distinct reasoning unit, even when humans would consider them implicit or automatic. This example reinforces a broader pattern we observe across the benchmark: models tend to overthink by breaking tasks into fine-grained, literal steps, while humans rely on holistic understanding and prior experience to simplify their reasoning. Humans can skip over obvious or familiar operations and focus on solving the puzzle efficiently. Another key difference is memory. Humans can leverage lifelong experience with similar puzzles and apply learned patterns without deliberation. In contrast, models reset their context at the beginning of each conversation and cannot reuse prior exposure unless explicitly prompted. They also lack common-sense filtering, treating all instructions and UI elements as equally important, which further inflates their reasoning depth estimates. This discrepancy highlights a core challenge in building effective agent systems: achieving human-like efficiency, intuition, and abstraction in multi-step reasoning. A robust benchmark must capture this behavioral gap.

## 3.3 Dataset Curation

As existing CAPTCHAs are for commercial use and not open-sourced, we cannot collect them online. Hence, we develop a data curation pipeline to construct the first open-sourced CAPTCHA dataset.

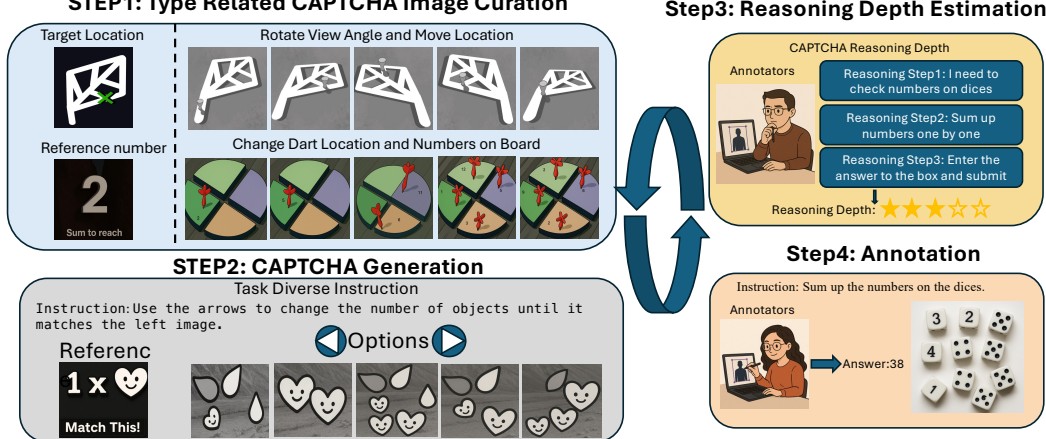

Figure 5: Open CaptchaWorld Date Curation Pipeline. **Step 1**: Curate diverse visual variations for each CAPTCHA type by modifying object positions, angles, and contextual cues. **Step 2**: Generate interactive tasks with human- or GPT-generated instructions tied to each image. **Step 3**: Estimate CAPTCHA Reasoning Depth. **Step 4**: Annotate final answers.

The images in our dataset are either generated by GPT-4o [30] or from human designers. To make data reliable, we use human annotators to create groundtruth and instructions. Fig. 5 demonstrates the pipeline to construct our dataset. We first brainstorm, search, and collect twenty CAPTCHA types. Then, for each type, the images are either generated from GPT-4o or designed by human artists. After we have all the images we need, we will design modern CAPTCHA tasks for each type which will need a multi-step, long horizon, and interactive actions (e.g., click, drag mouse cursor) task solving ability, notice that we do not test model's broad knowledge, so each CAPTCHA is actually could be solved by humans easily but hard for LLM Agents. Then, in step three, each type of CAPTCHA will be marked with our previously proposed *CAPTCHA Reasoning Depth* metrics by human annotators, these metrics and annotations can help us understand the different behaviors and misalignment of LLM Agents and humans when compared with their attempts to solve the CAPTCHAs. After all, the final ground truth solutions of CAPTCHAs will be annotated by annotators to make sure the ground truth is reliable, as humans can perform a 93.3% success rate in such a CAPTCHA environment, while LLM Agents are still far behind human performance. In addition, we show 20 examples from our Open CaptchaWorld in Fig. 2, covering all the types in dataset.

## 3.4 Multimodal Agents solve CAPTCHA

After curating the dataset and deploying our benchmark platform, we model the CAPTCHA-solving process of an agent as a finite-horizon partially observable Markov decision process (POMDP) [38], defined by the tuple:

$$\mathcal{M} = (\mathcal{S}, \mathcal{A}, \mathcal{O}, \mathcal{T}, \mathcal{Z}, R, \gamma) \tag{2}$$

where $\mathcal{S}$ is the latent environment state (e.g., CAPTCHA interface configuration), $\mathcal{A}$ is the action space (e.g., clicks, drags), $\mathcal{O}$ is the observation space (e.g., screenshots), $\mathcal{T}(s'|s, a)$ is the state transition probability, $\mathcal{Z}(o|s)$ is the observation function, $R(s, a)$ is the reward (success or failure), and $\gamma$ is the discount factor ( we set to 1 as we model CAPTCHA types equally) .

At each time step $t$, the agent receives an observation $o_t \in \mathcal{O}$ (e.g., screenshot), infers a belief state $b_t$, and selects an action $a_t \in \mathcal{A}$. The environment transitions to a new state $s_{t+1}$ and produces a new observation $o_{t+1}$. The agent aims to maximize the expected cumulative reward over the episode:

$$\mathbb{E}_\pi \left[ \sum_{t=0}^{T} \gamma^t R(s_t, a_t) \right] \tag{3}$$

## 4 Empirical Analysis

We systematically evaluate both base multimodal models and agent-based reasoning approaches on Open CaptchaWorld benchmark. To ensure fair comparisons, we adopt a unified experimental setup

with consistent prompting strategies and evaluation metrics applied across models and methods. In Section 4.1, we describe our evaluation protocol and implementation of Browser Use agents [26] equipped with different MLLM backbones. Section 4.2 presents the success rates of various models across all CAPTCHA types, highlighting the overall performance gap between humans and current agents. We then dive deeper in Section 4.3, conducting a fine-grained case study of success and failure patterns, categorized by task type and reasoning demand. Together, these analyses shed light on current limitations of multimodal agents and offer practical implications for future model design.

Table 1: Performance of different MLLM backbones within the Browser Use baseline agent on Open CaptchaWorld. Darker "⬛" indicates higher success rate@1 and darker "⬛" indicates higher cost. (See Appendix F for ablations on Agent Frameworks.)

| Solver Type | MLLM Backbone | Pass@1 (%) | Cost ($) |
|---|---|---|---|
| Human | – | **93.30** | - |
| Browser Use Agents | GPT-4o | 5.7 | 25.8 |
| | GPT-4.1 | 25.0 | 16.7 |
| | Claude-3.7-Sonnet | 20.0 | 18.7 |
| | Gemini2.5-Pro | 25.0 | 18.1 |
| | Openai-o3 | 40.0 | 66.4 |
| | Claude-3.5-Haiku | 15.0 | 9.3 |
| | Claude-3.5-Sonnet | 10.0 | 21.9 |
| | Openai-o1 | 5.0 | 94.6 |
| | DeepSeek-V3 | 20.0 | 7.3 |
| | Qwen2.5-VL-72b-Instruct | 11.0 | 13.9 |

## 4.1 Experimental Setup

We evaluate our benchmark in a zero-shot setting on Open CaptchaWorld. To better reflect real-world interaction needs and test powerful MLLM agents, we exclude traditional CAPTCHAs such as distorted text recognition or static image classification, as they can be solved by simple detection and classification models. All experiments are run in a web-based testing environment, where agents can perform multi-step actions like clicking, dragging, or typing. The CAPTCHAs are shown in a type-by-type sequence without repetition, ensuring that agents go through all puzzle types exactly once. We implement a Browser-Use Agent [26] system powered by different multimodal language models (MLLMs), including GPT-4o [30], GPT-4.1 (2025-04-14), Claude-3.7-Sonnet [4], Claude-3.5-Sonnet [2], Claude-3.5-Haiku [3], Gemini2.5-Pro [9], DeepSeek-V3 [43], and Openai-o3 (2025-04-16) [29]. These agents operate in a closed-loop setup: they receive screenshots of browser, reason about task, and issue actions step-by-step until they click final submit button. Moreover, the prompt we used to test Multimodal Agents is in Fig. 11.

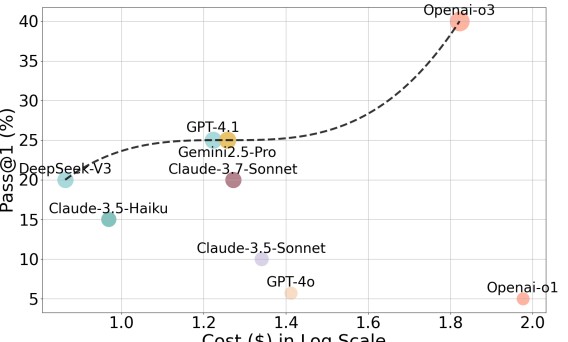

Figure 6: **Cost-performance trade-off among browser-use agents.** Each point represents a model, plotted by its evaluation cost (in log scale) and pass@1 success rate on Open CaptchaWorld. Openai-o3 achieves the highest success rate but incurs substantial cost, while models like Gemini2.5-Pro offer more favorable cost-effectiveness.

## 4.2 Success Rate of Multimodal Agents on Open CaptchaWorld

We evaluate our benchmark in a zero-shot setting using 20 types of modern CAPTCHA puzzles. To better reflect real-world interaction needs and test powerful MLLM agents, we exclude traditional CAPTCHA formats such as distorted text recognition or static image classification as they can be

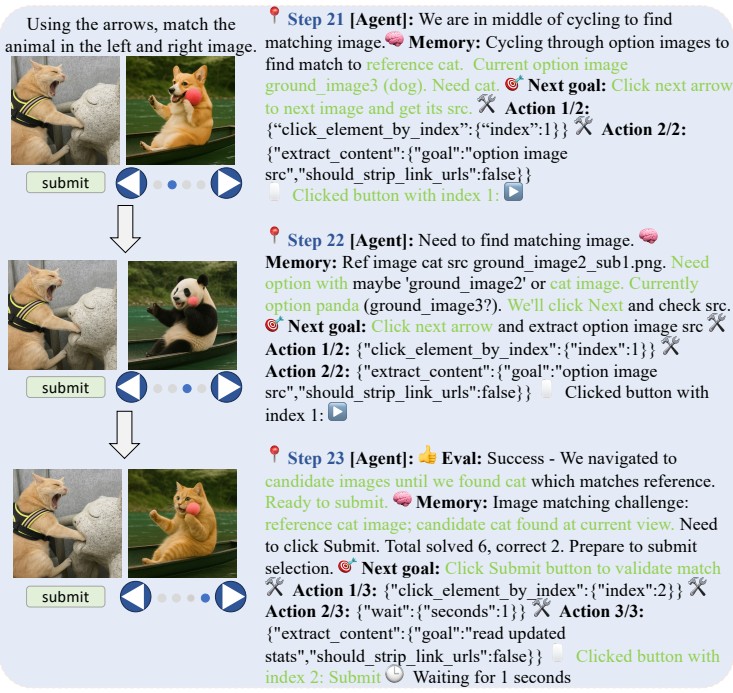

Figure 7: Step-by-step reasoning process of Openai-o3 in successfully solving Image Matching.

even solved by simple detection and classification models. All experiments are run in a web-based testing environment, where agents can perform multi-step actions like clicking, dragging, or typing. The CAPTCHAs are shown in a type-by-type sequence without repetition, ensuring that agents go through all puzzle types exactly once. We implement a Browser-Use Agent [26] system powered by different multimodal language models (MLLMs), including GPT-4o, GPT-4.1 (2025-04-14), Claude-3.7-Sonnet, Claude-3.5-Sonnet, Claude-3.5-Haiku, Gemini2.5-Pro, DeepSeek-V3, and Openai-o3 (2025-04-16). These agents operate in a closed-loop setup: they receive screenshots of browser, reason about task, and issue actions step-by-step until they click final submit button. Moreover, the prompt we used to test Multimodal Agents is in Fig. 11.

Table 5 presents the pass@1 success rate of various most advanced MLLM-powered browser-use agents on the Open CaptchaWorld benchmark. While human participants achieve an average success rate of 93.3%, all current models fall significantly short. The strongest performer, Openai-o3, reaches 40.0%, followed by GPT-4.1 and Gemini2.5-Pro at 25.0%. Other models, including Claude and GPT-4o variants, perform between 5.0% and 20.0%, with several showing near-random behavior on more complex tasks. In addition to performance, we also report the cost per evaluation episode in USD$, as shown in Table 5 and Fig 6. While Openai-o3 demonstrates the best success rate among agents, it also incurs a high cost of $66.4 per full CAPTCHA sequence, and GPT-4o and Claude-3-7-Sonnet show much lower performance at a moderate cost range. Notably, Openai-o1 yields the lowest success rate (5.0%) while being the most expensive ($94.6), making it the least cost-effective option. In contrast, models like DeepSeek-V3 and Claude-3-5-Haiku offer a more favorable balance of cost and performance, albeit at a relatively low accuracy.

These results highlight that model choice involves not only accuracy tradeoffs but also budget considerations, especially when deploying CAPTCHA-solving agents at scale. Cost-effective but robust agents remain an open challenge. Overall, the wide variance in both success rates and cost underscores the need for more efficient, reasoning-aligned MLLMs capable of performing real-world multi-step interactions with both accuracy and resource awareness.

## 4.3 Success and Failure Cases Analysis

As shown in Table 2 and Table 3, most models perform well on CAPTCHA types that rely primarily on basic visual perception, such as Image Matching, Object Match, Image Recognition, and especially Select Animal. Beyond the common types, OpenAI-o3 and Gemini2.5-Pro also perform well on

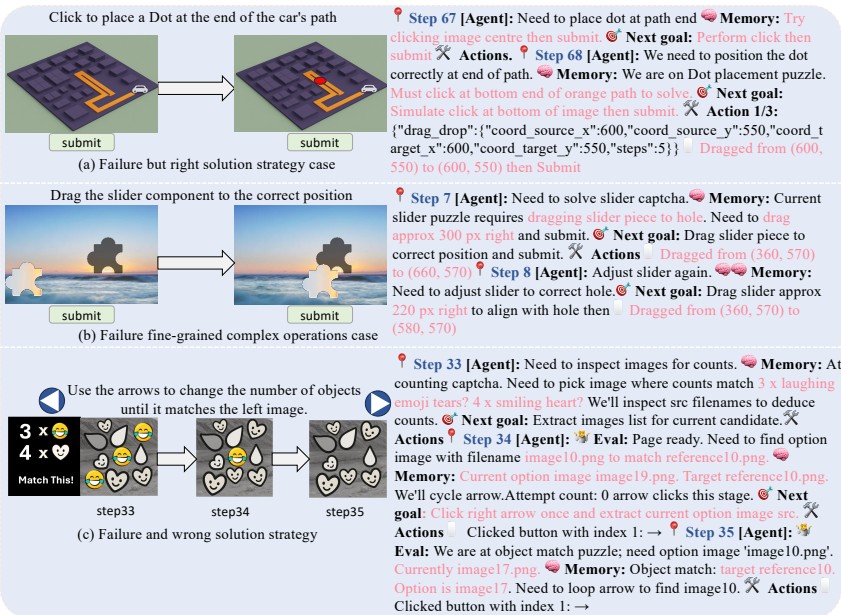

Figure 8: **Representative Failure of Openai-o3 Across Challenging CAPTCHA Types.** (a) Failure case with correct strategy but limited visual perception. (b) Failure case due to complex operational execution. (c) Failure case caused by misguided solution strategy based on irrelevant cues.

more challenging tasks such as Bingo. In addition, OpenAI-o1, GPT-4.1, and Gemini2.5-Pro exhibit strong performance on Dart Count, which involves arithmetic reasoning. Notably, Claude-3.7 stands out by uniquely succeeding on the Hold Button task, suggesting a higher level of operational control.

Given its strong overall performance and structured reasoning, we select Openai-o3 as a representative model to analyze across 20 CAPTCHA types, focusing on both successes and failures to assess its visual and cognitive abilities. Openai-o3 consistently solves tasks such as Object Match, Image Recognition, Select Animal, Image Matching, and Bingo. These tasks primarily depend on visual perception, object recognition, and basic reasoning, without requiring complex inference or interaction. Fig. 7 shows a successful example of o3 solving an Image Matching CAPTCHA: the model iteratively evaluates the current state, updates its memory, sets a goal, and cycles through candidate images until a match is found and submitted.

To better understand Openai-o3 model's limitations, we categorize its failure cases across challenging CAPTCHA types into three representative patterns, as illustrated in Fig. 8. These include: (a) failures where the model follows a generally correct solution strategy but lacks sufficient visual perception or spatial understanding, for instance, in the Place Dot task, it assumes the dot should be placed at the end of the path but repeatedly clicks near the center, missing the actual target; (b) failures involving fine-grained but complex operations, such as in the Slide Puzzle task, where the model understands the goal but fails to compute and execute the precise alignment needed; and (c) failures resulting from misguided strategies, such as in the Object Match task, where the model relies on image filenames or HTML text cues rather than visual analysis, leading to fundamentally incorrect solutions.

## 5   Conclusion

We introduce **Open CaptchaWorld**, the first open-source, web-based CAPTCHA benchmark for evaluating the interactive reasoning capabilities of multimodal LLM agents on diverse, modern CAPTCHA tasks. The benchmark targets a critical but underexplored challenge: enabling agents to perceive, reason, and act over multi-step tasks in dynamic web environments. Featuring 20 diverse CAPTCHA types and a novel CAPTCHA Reasoning Depth metric, it offers a task-agnostic measure of visual-cognitive difficulty. Through failure case analysis and observations of overthinking behavior, we expose key reasoning limitations in current agents. Open CaptchaWorld serves as a rigorous testbed for advancing more robust and human-aligned multimodal agents.

## Acknowledgments

This research is supported by the MBZUAI-WIS Joint Program for AI Research.

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

# Appendix of Open CaptchaWorld

**Contents**

## A   Limitation Statement

Despite the comprehensive scope of Open CaptchaWorld, there is a limitation. we evaluate agents in a zero-shot setting, which reflects current model generalization ability but may underestimate their potential under fine-tuning or with longer-term interaction memory. Future research can explore performance improvements through fine-tuning and memory-augmented learning. Specifically, incorporating few-shot demonstrations or reinforcement learning from interaction history could enable agents to better adapt to specific CAPTCHA types or recurring visual patterns. Moreover, integrating longer-term memory modules may allow models to accumulate task-solving strategies over multiple interactions, bridging the gap between isolated inference and real-world usage scenarios where adaptation and learning over time are essential.

## B   Broader Impacts

**Positive Societal Impacts.** Our work introduces *Open CaptchaWorld*, the first open-source benchmark for evaluating LLM-based multimodal agents on interactive CAPTCHA tasks. This fills a crucial gap in the evaluation of AI agents by focusing on a realistic challenge that frequently arises in practical web environments—human verification. By highlighting current limitations in agent robustness, reasoning efficiency, and interaction capabilities, our benchmark provides the research community with a principled framework to develop more capable, safe, and trustworthy AI systems. It also promotes transparency and reproducibility in a domain that has often relied on closed-source, commercial CAPTCHA datasets. Ultimately, this research could accelerate the deployment of more effective accessibility agents for users with visual or motor impairments, who currently struggle with CAPTCHAs.

**Potential Negative Societal Impacts.** A potential concern is that progress in solving CAPTCHAs with LLM agents may lower the barrier for malicious automation, such as bot-driven exploitation of web services, fake account creation, or scraping of private content. While our benchmark is designed for academic purposes and emphasizes ethical data use, we acknowledge that improved CAPTCHA-solving capabilities may also be misused. Moreover, automating CAPTCHA solutions could undermine existing web security mechanisms, especially if deployed irresponsibly. To mitigate such risks, we recommend that any system developed using insights from our benchmark be gated, logged, and constrained within authorized use cases. Finally, while we employ human annotators ethically and compensate them fairly, continued reliance on human labor in data annotation pipelines raises broader questions around labor rights, scalability, and annotator fatigue.

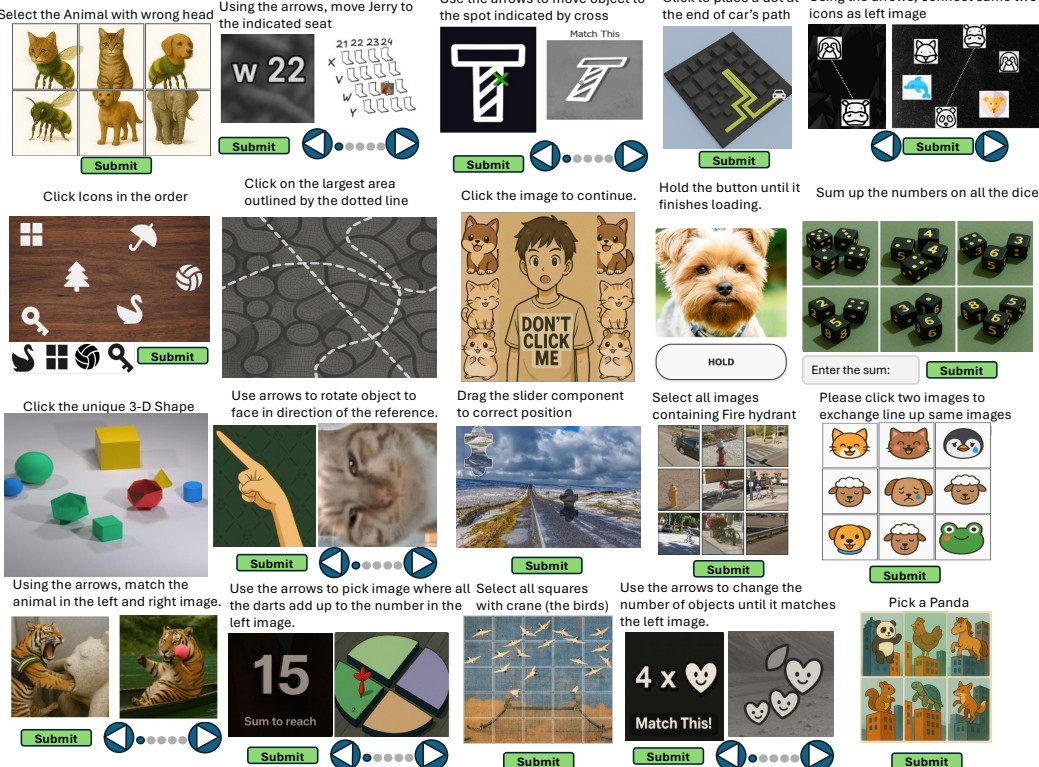

Figure 9: **More Examples of Open CaptchaWorld.**

## C  More Examples from Open CaptchaWorld

Here we provide more examples of CAPTCHAs in our Open CaptchaWorld Benchmark, as shown in Figure 9. Notice that all the images for each CAPTCHA are not repeated.

## D  MLLM Models Performance Analysis on Different CAPTCHA Types

Table 2 presents a capability support matrix that summarizes whether each multimodal agent successfully solved at least one instance of each CAPTCHA type in our benchmark. A "✓" indicates that the model demonstrated at least partial success on that type, while "✗" indicates complete failure across all test instances. This table helps visualize the distribution of strengths and weaknesses among different MLLM agents. We observe that certain tasks—such as *Image Recognition*, *Image Matching*, and *Select Animal*—are universally solved by nearly all models, suggesting they rely primarily on basic visual grounding or object recognition. In contrast, tasks requiring spatial manipulation (*Slide Puzzle*), counting (*Dice Count*), dynamic control (*Hold Button*), or the ability to interpret visual instructions and extract relevant cues from images (*Pick Area*, *Misleading Click*) remain unsolved by almost all the models.

Notably, Claude3.7-Sonnet show isolated strengths—for instance, uniquely solving *Hold Button* task—indicating variation in architectural strengths or alignment training. This breakdown reinforces that existing MLLM agents exhibit significant variance in cross-task generalization and often struggle with interaction-heavy or arithmetic-based challenges. The table serves as a diagnostic tool for future model benchmarking and agent specialization analysis.

Table 3 presents the Pass@1 success rates (%) of each evaluated model across 20 different CAPTCHA types. Each CAPTCHA type contains more than ten samples to ensure statistical validity. The reported scores are averaged over three runs, with standard deviations indicated to reflect consistency and reliability across attempts. From another perspective, large variances in certain cases suggest that models are learning to solve specific tasks more effectively over repeated trials, possibly leveraging

Table 2: Support of different models on various types of CAPTCHA tasks.

| | Openai-o3 | Openai-o1 | GPT-4.1 | GPT-4o | Gemini2.5-Pro | Claude3.7-Sonnet | Claude3.5-Haiku | Claude3.5-Sonnet | DeepSeek-V3 |
|---|---|---|---|---|---|---|---|---|---|
| Dice_Count | ✗ | ✗ | ✗ | ✓ | ✗ | ✗ | ✗ | ✗ | ✗ |
| Geometry_Click | ✓ | ✗ | ✗ | ✗ | ✗ | ✗ | ✗ | ✗ | ✗ |
| Rotation_Match | ✓ | ✓ | ✓ | ✓ | ✓ | ✓ | ✓ | ✓ | ✓ |
| Slide_Puzzle | ✗ | ✗ | ✗ | ✗ | ✗ | ✗ | ✗ | ✗ | ✗ |
| Unusual_Detection | ✓ | ✓ | ✓ | ✓ | ✓ | ✗ | ✓ | ✓ | ✓ |
| Image_Recognition | ✓ | ✓ | ✓ | ✓ | ✓ | ✓ | ✓ | ✓ | ✗ |
| Bingo | ✓ | ✓ | ✓ | ✗ | ✓ | ✗ | ✗ | ✗ | ✗ |
| Image_Matching | ✓ | ✓ | ✓ | ✓ | ✓ | ✓ | ✓ | ✓ | ✓ |
| Patch_Select | ✗ | ✓ | ✓ | ✓ | ✗ | ✗ | ✗ | ✗ | ✗ |
| Dart_Count | ✓ | ✓ | ✓ | ✓ | ✓ | ✓ | ✓ | ✓ | ✓ |
| Object_Match | ✓ | ✓ | ✓ | ✓ | ✓ | ✓ | ✓ | ✓ | ✓ |
| Select_Animal | ✓ | ✓ | ✓ | ✓ | ✓ | ✓ | ✓ | ✓ | ✓ |
| Coordinates | ✓ | ✓ | ✓ | ✓ | ✓ | ✓ | ✓ | ✓ | ✓ |
| Path_Finder | ✓ | ✓ | ✓ | ✓ | ✓ | ✓ | ✓ | ✓ | ✓ |
| Connect_icon | ✓ | ✓ | ✓ | ✓ | ✓ | ✓ | ✓ | ✓ | ✓ |
| Click_Order | ✗ | ✗ | ✗ | ✗ | ✗ | ✗ | ✗ | ✗ | ✗ |
| Hold_Button | ✗ | ✗ | ✗ | ✗ | ✗ | ✓ | ✗ | ✗ | ✗ |
| Misleading_Click | ✗ | ✗ | ✗ | ✗ | ✗ | ✗ | ✗ | ✗ | ✗ |
| Pick_Area | ✗ | ✗ | ✗ | ✗ | ✗ | ✗ | ✗ | ✗ | ✗ |

experience accumulated during interaction. This table provides a comprehensive view of each model's capabilities on a wide spectrum of visual perception, reasoning, and interaction-intensive challenges.

## E  Reasoning Depth Annotation Guidelines

To estimate the **Reasoning Depth** of a CAPTCHA puzzle, we define a checklist of atomic reasoning and interaction steps that a human must perform. Each step corresponds to a discrete visual, cognitive, motor, or state-transition operation. A CAPTCHA's total reasoning depth is computed by counting how many of these atomic steps are required to solve it correctly. Each satisfied atomic step contributes a depth of +1.

Annotators are instructed to use the following table as a reference. For every puzzle analyzed, they should determine which of the atomic steps are involved, and report the total reasoning depth accordingly. For transparency, all annotations must be accompanied by justifications that cite specific steps from the table.

Table 3: Pass@1 success rate (%) of different models on various types of CAPTCHA tasks.

| | Openai-o3 | Openai-o1 | GPT-4.1 | GPT-4o | Gemini2.5-Pro | Claude3.7-Sonnet | Claude3.5-Haiku | Claude3.5-Sonnet | DeepSeek-V3 |
|---|---|---|---|---|---|---|---|---|---|
| Dice_Count | 0.0 ± 0.0 | 0.0 ± 0.0 | 0.0 ± 0.0 | 3.3 ± 4.7 | 0.0 ± 0.0 | 0.0 ± 0.0 | 0.0 ± 0.0 | 0.0 ± 0.0 | 0.0 ± 0.0 |
| Geometry_Click | 6.7 ± 4.7 | 0.0 ± 0.0 | 0.0 ± 0.0 | 0.0 ± 0.0 | 0.0 ± 0.0 | 0.0 ± 0.0 | 0.0 ± 0.0 | 0.0 ± 0.0 | 0.0 ± 0.0 |
| Rotation_Match | 6.7 ± 4.7 | 10.0 ± 8.2 | 3.3 ± 4.7 | 6.7 ± 9.4 | 20.0 ± 0.0 | 13.3 ± 4.7 | 13.3 ± 12.5 | 6.7 ± 4.7 | 13.3 ± 9.4 |
| Slide_Puzzle | 0.0 ± 0.0 | 0.0 ± 0.0 | 0.0 ± 0.0 | 0.0 ± 0.0 | 0.0 ± 0.0 | 0.0 ± 0.0 | 0.0 ± 0.0 | 0.0 ± 0.0 | 0.0 ± 0.0 |
| Unusual_Detection | 6.7 ± 4.7 | 6.7 ± 4.7 | 13.3 ± 4.7 | 10.0 ± 8.2 | 13.3 ± 4.7 | 0.0 ± 0.0 | 13.3 ± 4.7 | 16.7 ± 12.5 | 3.3 ± 4.7 |
| Image_Recognition | 23.3 ± 4.7 | 23.3 ± 4.7 | 23.3 ± 4.7 | 10.0 ± 8.2 | 23.3 ± 18.9 | 3.3 ± 4.7 | 10.0 ± 8.2 | 40.0 ± 0.0 | 0.0 ± 0.0 |
| Bingo | 60.0 ± 0.0 | 43.3 ± 4.7 | 6.7 ± 4.7 | 0.0 ± 0.0 | 56.7 ± 12.5 | 0.0 ± 0.0 | 0.0 ± 0.0 | 0.0 ± 0.0 | 0.0 ± 0.0 |
| Image_Matching | 43.3 ± 17.0 | 33.3 ± 12.5 | 53.3 ± 17.0 | 16.7 ± 9.4 | 23.3 ± 12.5 | 60.0 ± 8.2 | 30.0 ± 0.0 | 56.7 ± 9.4 | 20.0 ± 0.0 |
| Patch_Select | 0.0 ± 0.0 | 3.3 ± 4.7 | 3.3 ± 4.7 | 3.3 ± 4.7 | 0.0 ± 0.0 | 0.0 ± 0.0 | 0.0 ± 0.0 | 0.0 ± 0.0 | 0.0 ± 0.0 |
| Dart_Count | 13.3 ± 4.7 | 43.3 ± 9.4 | 50.0 ± 0.0 | 13.3 ± 4.7 | 43.3 ± 9.4 | 40.0 ± 8.2 | 13.3 ± 9.4 | 46.7 ± 9.4 | 6.7 ± 9.4 |
| Object_Match | 26.7 ± 4.7 | 20.0 ± 14.1 | 26.7 ± 12.5 | 23.3 ± 4.7 | 33.3 ± 12.5 | 46.7 ± 17.0 | 6.7 ± 4.7 | 23.3 ± 17.0 | 10.0 ± 0.0 |
| Select_Animal | 100.0 ± 0.0 | 100.0 ± 0.0 | 83.3 ± 4.7 | 93.3 ± 4.7 | 100.0 ± 0.0 | 100.0 ± 0.0 | 96.7 ± 4.7 | 100.0 ± 0.0 | 20.0 ± 0.0 |
| Coordinates | 10.0 ± 0.0 | 16.7 ± 4.7 | 23.3 ± 4.7 | 23.3 ± 4.7 | 26.7 ± 20.5 | 10.0 ± 0.0 | 20.0 ± 8.2 | 16.7 ± 4.7 | 23.3 ± 4.7 |
| Path_Finder | 16.7 ± 4.7 | 23.3 ± 4.7 | 26.7 ± 9.4 | 30.0 ± 0.0 | 30.0 ± 21.6 | 10.0 ± 8.2 | 20.0 ± 8.2 | 20.0 ± 0.0 | 23.3 ± 4.7 |
| Place_Dot | 0.0 ± 0.0 | 0.0 ± 0.0 | 0.0 ± 0.0 | 0.0 ± 0.0 | 0.0 ± 0.0 | 0.0 ± 0.0 | 0.0 ± 0.0 | 0.0 ± 0.0 | 0.0 ± 0.0 |
| Connect_icon | 10.0 ± 0.0 | 10.0 ± 0.0 | 16.7 ± 2.4 | 18.3 ± 2.4 | 13.3 ± 4.7 | 30.0 ± 8.2 | 13.3 ± 2.4 | 11.7 ± 6.2 | 36.7 ± 4.7 |
| Click_Order | 0.0 ± 0.0 | 0.0 ± 0.0 | 0.0 ± 0.0 | 0.0 ± 0.0 | 0.0 ± 0.0 | 0.0 ± 0.0 | 0.0 ± 0.0 | 0.0 ± 0.0 | 0.0 ± 0.0 |
| Hold_Button | 0.0 ± 0.0 | 0.0 ± 0.0 | 0.0 ± 0.0 | 0.0 ± 0.0 | 0.0 ± 0.0 | 100.0 ± 0.0 | 0.0 ± 0.0 | 0.0 ± 0.0 | 0.0 ± 0.0 |
| Misleading_Click | 0.0 ± 0.0 | 0.0 ± 0.0 | 0.0 ± 0.0 | 0.0 ± 0.0 | 0.0 ± 0.0 | 0.0 ± 0.0 | 0.0 ± 0.0 | 0.0 ± 0.0 | 0.0 ± 0.0 |
| Pick_Area | 0.0 ± 0.0 | 0.0 ± 0.0 | 0.0 ± 0.0 | 0.0 ± 0.0 | 0.0 ± 0.0 | 0.0 ± 0.0 | 0.0 ± 0.0 | 0.0 ± 0.0 | 0.0 ± 0.0 |

**Prompt to Estimate CAPTCHA Reasoning Depth**

To estimate the reasoning depth of a CAPTCHA puzzle, use the following rules as checklist: {Rules}. The goal is to assess how many discrete reasoning or interaction steps a human would need to solve the puzzle. Use the provided checklist as a reference, but do not rigidly count checklist items. Instead, reflect on the actual sequence of cognitive and motor steps needed to reach the solution. For each estimated depth, clearly explain your reasoning.

Figure 10: Prompt for estimating CAPTCHA Reasoning Depth.

**Prompt to Test Browser Use Agents on Open CaptchaWorld**

You are an autonomous CAPTCHA-solver for the **Open CaptchaWorld** webpage. Go to http://localhost:5001/ and solve the CAPTCHA challenges as many as you can. Notice, You may need to click some buttons to solve the captcha.

Figure 11: **Prompt to Browser Use Agents for testing on Open CaptchaWorld.**

Table 4: Checklist of Atomic Steps for Reasoning Depth Estimation

| Category | Atomic Step Description |
| --- | --- |
| **Visual (V)** | Locate a single target object class |
| | Read an entire multi-character CAPTCHA string |
| | Detect orientation of one jigsaw tab |
| | Identify a color-coded region |
| | Recognize a specific symbol or emoji |
| | Count objects in a scene |
| | Spot the difference between two panels |
| | Read numeric code displayed on a dial |
| | Interpret a legend or map key |
| | Detect newly revealed hint after a state change |
| **Cognitive (C)** | Decide a subset satisfying a logical rule |
| | Plan the order of assembling pieces |
| | Infer a hidden sorting principle |
| | Translate a textual instruction into an action set |
| | Choose the optimal path in a maze |
| | Determine the required rotation angle before acting |
| | Resolve ambiguity between visually similar targets |
| | Memorize a short cue for later recall |
| | Select the correct tool among many options |
| | Apply elimination logic to narrow down choices |
| **Motor (M)** | Single left-click on a target |
| | Bulk-select multiple tiles after a single decision |
| | Drag-and-drop one piece (grab → release) |
| | Continuous slider movement to a target position |
| | Rotate a dial or knob in one continuous motion |
| | Type a full code in one uninterrupted entry |
| | Draw a single straight line |
| | Resize a bounding box once |
| | Check or uncheck a checkbox |
| | Press-and-hold a button until success |
| **State Reveal (V)** | Observe the puzzle state after an automatic change |

# F    Ablations on Different Agent Frameworks

Table 5: Performance of different popular web-agent / multi-agent framework using GPT-4o as backbone on Open CaptchaWorld. Darker "■" indicates higher success rate@1 and darker "■" indicates higher cost.

| Agent Framework | MLLM Backbone | Pass@1 (%) |
|---|---|---|
| Human | – | **93.30** |
| Browser Use Agents | GPT-4o | 5.7 |
| SeeAct | GPT-4o | 7.0 |
| WebVoyager | GPT-4o | 9.0 |
| OWL | GPT-4o | 10.0 |

