# OpenReview forum: "Open CaptchaWorld: A Comprehensive Web-based Platform for Testing and Benchmarking Multimodal LLM Agents"
_NeurIPS.cc/2025/Datasets_and_Benchmarks_Track — NeurIPS 2025 Datasets and Benchmarks Track poster_

### Official Review · Reviewer_uzRf · 2025-06-29

**Rating:** 5
**Confidence:** 4

**Summary:**

Authors propose Open CaptchaWorld, the first open-source, large-scale, and long-term maintaining CAPTCHA benchmark for evaluating interactive multimodal agents using MLLMs. Authors introduce CAPTCHA Reasoning Depth, a task-agnostic complexity measure capturing the multi-step reasoning burden of visual interaction puzzles. Authors build a real web-based testing platform and systematically evaluate state-of-the-art models in zero-shot settings, revealing large performance gaps compared to humans. Authors provide insights into agent failure cases such as overthinking, over-segmentation and interface misunderstanding.

**Dataset Code Accessibility:**

Yes

**Dataset Code Comments:**

The code looks correct, the documentation is well done.

**Ethical Considerations:**

No, there are no or only very minor ethics concerns

**Final Justification:**

My comments were correctly taken into account. I recommend the article for acceptance.

**Limitations Weaknesses:**

1. The main limitation of the work is the dataset scale. While diverse, 225 CAPTCHAs contained in the Open CaptchaWorld dataset may lack the scale needed for robust statistical analysis, especially for less common types. Expanding the dataset could improve generalizability.

2. The authors are advised to add the capabilities of the Qwen2.5-VL multimodal model [1] to the performance comparison of modern models on the developed dataset.
[1] Bai S, Chen K, Liu X, Wang J, Ge W, Song S, Dang K, Wang P, Wang S, Tang J, Zhong H. Qwen2. 5-vl technical report. arXiv preprint arXiv:2502.13923. 2025

3. Punctuation marks are missing at the end of all formulas.

**Strengths Contributions:**

* Open CaptchaWorld benchmark fills a gap by focusing on interactive CAPTCHAs, which are systematically excluded in existing benchmarks. This addresses a real-world bottleneck for web agents.
* The proposed CAPTCHA Reasoning Depth metric provides a task-agnostic measure of complexity, grounded in human annotation studies.
* The paper evaluates 9 MLLM agents in a zero-shot, browser-based POMDP environment, revealing stark performance disparities. The authors also demonstrate the analysis of failure modes.
* The benchmark’s design and open-source release (Hugging Face platform) facilitate reproducibility and community engagement.
* The paper is well-structured, with clear detailed figures and thorough appendices.

---

> ### Author Rebuttal · Authors · 2025-07-30
>
> We thank the reviewer for the insightful comments and constructive suggestions. We have carefully considered all the points raised and have revised the manuscript accordingly. Below, we provide a point-by-point response to each comment.
>
>
> >**W1. The main limitation of the work is the dataset scale. While diverse, 225 CAPTCHAs contained in the Open CaptchaWorld dataset may lack the scale needed for robust statistical analysis, especially for less common types. Expanding the dataset could improve generalizability.**
>
> We appreciate the reviewer's valuable suggestion. We have doubled the total dataset size now, and as this is a long-term project, further scaling will be implemented.
>
>
>
> >**W2. Add the capabilities of the Qwen2.5-VL multimodal model to the performance comparison of modern models on the developed dataset.**
>
> We truly appreciate this comment. Here is the result of Qwen2.5-VL-72b-Instruct [1] and we will add to our revised paper. For a fair comparison of the cost, we use Alibaba Cloud's DASHSCOPE API instead of inference with open-sourced weights:
>
>
> | MLLM Backbone   | Pass@1(%)| Cost($) |
> |---------|-----------:|----------:|
> | Qwen2.5-VL-72b-Instruct [1] | 11.0%        | 13.9      |
>
>
> >**W3. Punctuation marks are missing at the end of all formulas.**
>
> Thanks for your careful review. We will add the punctuation marks in the revised version.
>
> [1] Bai S, Chen K, Liu X, Wang J, Ge W, Song S, Dang K, Wang P, Wang S, Tang J, Zhong H. Qwen2. 5-vl technical report. arXiv preprint arXiv:2502.13923. 2025.

---

> > ### Comment · Reviewer_uzRf · 2025-08-07
> >
> > My comments were correctly taken into account. I recommend the article for acceptance.

---

### Official Review · Reviewer_AAv3 · 2025-07-02

**Rating:** 5
**Confidence:** 4

**Summary:**

The author proposed the Open Catcha dataset, aimed at detecting the ability of MLLM agents to solve Captchas, with the expectation of deploying agents in real web environments. The author introduced a new metric called Reasoning Depth, which classifies tasks from the perspectives of difficulty and cognition, and compared the calculation processes of humans and models for this metric. The results show that current advanced MLLMs still have a significant gap in completing Captcha tasks, highlighting a critical issue that must be addressed for deploying agents in real web environments.

**Dataset Code Accessibility:**

Yes

**Dataset Code Comments:**

I can find the author's dataset and have tested the huggingface space they provided.

**Ethical Considerations:**

No, there are no or only very minor ethics concerns

**Final Justification:**

I have increased my score and recommend accepting this paper.

**Limitations Weaknesses:**

1. **CAPTCHA Reasoning Depth:** The determination of this metric is kind of subjective and lacks clarity in its definition. For instance, in the example shown in Figure 4, doesn't each click count as a reasoning step? I noticed that when defining depth, you mentioned that “each step involves interpreting visual content, planning a subgoal, or executing a discrete interaction.”

2. During the experimental testing, how does each MLLM interact with the computer? Specifically, how is their textual output mapped to computer operations? Could different processing methods potentially affect the results?

**Strengths Contributions:**

1. Captcha is indeed a challenge that web agents must overcome. The author keenly identified this difficulty and created corresponding evaluation benchmarks, filling a gap in the assessment of web intelligent agents.

2. The significant gap between MLLM and humans on this benchmark shows that this is an area that has not yet been solved, even though MLLM already possesses strong visual parsing capabilities.

3. Detailed case analysis.

---

> ### Author Rebuttal · Authors · 2025-07-30
>
> We are grateful to the reviewer for the time and effort in providing valuable feedback. These comments have been instrumental in strengthening our manuscript. We have addressed all suggestions in the revised version, with detailed responses provided below.
>
> >**W1. CAPTCHA Reasoning Depth: The determination of this metric is kind of subjective and lacks clarity in its definition. For instance, in the example shown in Figure 4, doesn't each click count as a reasoning step? I noticed that when defining depth, you mentioned that “each step involves interpreting visual content, planning a subgoal, or executing a discrete interaction.”**
>
> We thank the reviewer for raising this important question, which helps us clarify the definition of CAPTCHA Reasoning Depth and how it is applied. The reviewer's observation that a "step" can be more than a single physical action is correct and is a key feature of our metric.
>
> The core idea is that a "step" corresponds to a distinct decision-making point or a cognitive subgoal, not necessarily every individual motor action. The definition "each step involves interpreting visual content, planning a subgoal, or executing a discrete interaction" was chosen carefully, as a block of related interactions performed to satisfy a single, pre-planned subgoal is counted as one step. In the example from Figure 4 ("Click Icons in the order"), the human annotator decomposes the task into three cognitive subgoals: planning the sequence, executing the clicks as one "chunked" interaction block, and finally, submitting the answer.
>
> Related to the question "Doesn't each click count as a reasoning step?".  For the Agent model, it often does. As shown in the O3 response, the model explicitly breaks this down further into "execute those clicks one by one", leading to a higher depth score. This distinction is formalized in our annotation guidelines (Table 4), which explicitly differentiate between a "Single left-click on a target" and a "Bulk-select multiple tiles after a single decision". To be specific, the "Click Icons in order" task, for a human, falls under the second definition, where a single decision precedes a chain of motor actions counted as one execution block. This proves our method is not subjective but follows a predefined heuristic. By defining a step around cognitive subgoals, our metric effectively quantifies task complexity and reveals the key reasoning gap between human intuition and the more literal, over-segmented process of current MLLM agents.
>
> >**W2. During the experimental testing, how does each MLLM interact with the computer? Specifically, how is their textual output mapped to computer operations? Could different processing methods potentially affect the results?**
>
>
> We thank the reviewer for the question. The process is as follows:
>
> At each step, the MLLM agent receives the current observation from the browser, which is a screenshot of the web page.
> The MLLM then reasons about the task and generates a textual output. This output is not free-form natural language, but a structured command in a JSON-like format that specifies the exact action to be taken (e.g., {"click_element_by_index":{"index": 11}} or {"drag_drop": ...}). Examples of these precise action outputs can be seen in our failure analysis in Figures 7 and 8. The Browser-Use framework then parses this structured command and executes the corresponding deterministic operation in the browser.
>
> As for the question "if different processing methods could affect the results", the answer is yes, they absolutely could. This is precisely why we employed a single, consistent framework for all tested models. By keeping the action space and the parsing mechanism constant, we ensure that the observed performance differences are attributable to the reasoning and instruction-following capabilities of the MLLM backbones themselves, rather than variations in the interaction framework. While a different agent framework might yield different absolute success rates, our unified approach is the correct methodology for isolating and fairly comparing the core capabilities of the models, which is the central objective of our study.
>
> We have conducted experiments to compare several Web Agent Frameworks, which demonstrate the differences as shown in the following table:
>
> Agent Framework | Pass@1(%)|
> |---------|-----------:|
> |Browser-Use|5.7%|
> |SeeAct [1]|7.0%|
> |WebVoyager [2]|9.0%|
> |OWL [3]|10.0%|
>
> [1] Zheng B, Gou B, Kil J, Sun H, Su Y. Gpt-4v (ision) is a generalist web agent, if grounded. arXiv preprint arXiv:2401.01614. 2024.
>
> [2] He H, Yao W, Ma K, Yu W, Dai Y, Zhang H, Lan Z, Yu D. Webvoyager: Building an end-to-end web agent with large multimodal models. arXiv preprint arXiv:2401.13919. 2024.
>
> [3] Hu M, Zhou Y, Fan W, Nie Y, Xia B, Sun T, Ye Z, Jin Z, Li Y, Chen Q, Zhang Z. Owl: Optimized workforce learning for general multi-agent assistance in real-world task automation. arXiv preprint arXiv:2505.23885. 2025.

---

### Official Review · Reviewer_Hkz5 · 2025-07-03

**Rating:** 4
**Confidence:** 3

**Summary:**

The authors collect a wide range of 20 Captcha types and construct a comprehensive benchmark, too mainly test MLLMs' and agents' ability to solve multi-step and long-term visual puzzles. A new metric named reasoning depth is provided to evaluate the difficulty of each puzzle. In experiment part, the authors evaluate different MLLMs under the browser use agent framework.

**Dataset Code Accessibility:**

Yes

**Dataset Code Comments:**

Dataset provided in huggingface
Code provided too

**Ethical Considerations:**

No, there are no or only very minor ethics concerns

**Final Justification:**

The authors have clarified my questions. I keep a positive attitude towards this paper.

**Limitations Weaknesses:**

- Though it is the first Captcha benchmark as the authors claim. What the difference between Captcha and other vision agent benchmarks like VisualWebArena, as they both requires vision recognition, long-term multi-step action.
- I am confused about the atomic action in Tab. 4. Does actions like "locate a ..." involve external tools? If not, how can you decompose depth from a consecutive reasoning process from human or MLLM?
- The authors mainly focus on a single agent type "Browser use" with different underlying MLLMs. Is there any other web agent frameworks other than browser use?

**Strengths Contributions:**

- The task of solving Captcha is novel, and the benchmark includes a wide range of Captcha types.
- The authors provide a comprehensive evaluation on a wide range of MLLMs.

---

> ### Author Rebuttal · Authors · 2025-07-30
>
> We sincerely thank the reviewer for the constructive comments, which are helpful in improving the quality of our paper. We will carefully revise accordingly and include the suggested changes in the revised manuscript. In the following, we detail our responses to each of the questions raised.
>
>
>
> >**W1. Though it is the first Captcha benchmark as the authors claim. What the difference between Captcha and other vision agent benchmarks like VisualWebArena, as they both requires vision recognition, long-term multi-step action.**
>
> We truly appreciate your comment. We wish to clarify the fundamental differences between VisualWebArena [1] and OpenCaptchaWorld in the purpose, task nature, and the specific capabilities evaluate, which makes our benchmark a complementary and necessary addition to the field as follows:
>
>
> (1) VisualWebArena [1] evaluates an agent's ability to perform realistic, user-centric tasks on websites designed for human interaction (e.g., shopping, browsing Reddit). In contrast, our Open CaptchaWorld evaluates an agent's ability to solve captchas that are explicitly and adversarially designed to distinguish humans from bots and block automation. In VisualWebArena, the main challenge lies in achieving web usage tasks, whereas in Open CaptchaWorld, it is overcoming a security gatekeeper that blocks the agent’s progress. Moreover, Unlike knowledge-core tasks, the captchas require the model to deduce novel rules and logic purely from the immediate visual context, providing a more rigorous test of fluid and abstract reasoning rather than recall.
>
> (2) The reasoning required to solve a captcha is distinctive, often demanding non-literal thinking, the recognition of intentional image alterations, and an understanding of ‘trick’ logic that is typically absent from standard web tasks. By contrast, tasks in VisualWebArena are visually grounded but functionally straightforward (e.g., ‘Buy the cheapest color photo printer’). For example, tasks in Open CaptchaWorld require agents to handle challenges like "Select the Animal with wrong head" or "Click on the largest area outlined by the dotted line", which demand a different, more robust form of perception and reasoning against confusion.
>
> To be specific, captchas are like intelligient puzzles that are intentionally crafted to be simple for humans but difficult for models, specifically targeting the gap in "common sense" reasoning. Challenges like "Select the Animal with wrong head" demand an intuitive understanding of absurdity, a key facet of human intelligence. Furthermore, many of our puzzles test for robustness against intentionally deceptive information and require fine-grained spatial logic, such as in "Drag the slider component to correct position," which evaluates visuomotor control in a way that standard web navigation does not. The testing of visuomotor control is underexplored in other benchmarks like web-based tasks benchmark (**VisualWebArena [1]** ) or visual-puzzle based (**VisualPuzzle [2]**), this underestimate will cause inaccurate operations.
>
> For instance, recent web agents increasingly utilize visual grounding models as their localization tools, such as "**Set-of-Marks [3]**" (SoM) method prominently featured in VisualWebArena.  These tools typically operate by segmenting a webpage into a finite set of interactable elements, and agent's task is reduced to selecting which element to interact with. However, action itself is often simplified; a 'click' command, for example, will target  geometric center of selected element's bounding box. This approach is sufficient for most standard web interfaces but fails on challenges requiring more nuanced spatial interaction. Many of our captchas expose this limitation directly. A puzzle might require an agent to click "upper right" corner of an image or place a dot at precise "end of car's path". In these scenarios, an agent that can only identify the correct image element but is incapable of clicking a specific point within it is destined to fail, revealing a critical gap in fine-grained visuomotor control that our benchmark is uniquely designed to evaluate.
> Hence, In summary, the focus on non-literal thinking, cognitive resilience, and spatial understanding against obfuscation makes our benchmark a unique and necessary tool for measuring the true intelligence of a model.
>
> (3) As we state in our paper, most existing agent benchmarks, including the WebArena framework that VisualWebArena is built upon, systematically filter out or avoid websites containing captchas. This creates a critical gap, as captchas remain one of the major obstacles preventing agents from completing real-world tasks. Open CaptchaWorld is therefore not a competing benchmark, but a complementary one, specifically designed to address this underexplored bottleneck. An agent that achieves a high score on VisualWebArena may still perform poorly on Open CaptchaWorld and  more importantly, will struggle in many real-world scenarios if it cannot bypass captchas at crucial steps like login or checkout.
>
>
> >**W2. I am confused about the atomic action in Tab. 4. Does actions like "locate a ..." involve external tools? If not, how can you decompose depth from a consecutive reasoning process from human or MLLM?**
>
> We thank the reviewer for this perceptive question regarding the decomposition of "CAPTCHA Reasoning Depth". We would like to clarify our methodology. First, whether involves the external tools depends on the Agent Frameworks, for the browser-use framework we tested, the "Set-of-Marks [3]" tool will be used to help locate.
>
> To decompose and quantify the internal cognitive process into discrete steps, we employed a standard methodology as detailed in Section 3.2, "we conducted a human annotation study where participants solved sample puzzles while verbally decomposing their thought process into atomic reasoning steps, guided by a set of heuristic rules (Table 4)". This "think-aloud" protocol makes the otherwise internal reasoning process observable and reportable. The process is not arbitrary. It is structured in two key ways: The checklist in Table 4 provides a consistent, predefined set of categories that guide the annotators in how they label their thought process. As stated in our paper, we "measured inter-annotator agreement to ensure consistency", confirming that this decomposition can be performed reliably across different human annotators.
>
>
>
>
> >**W3. The authors mainly focus on a single agent type "Browser use" with different underlying MLLMs. Is there any other web agent frameworks other than browser use?**
>
> Yes, there are other web agent frameworks such as SeeAct [4], WebVoyager [5], and OWL [6]. While our study focuses on the “Browser Use” agent type for its controlled and reproducible setup across MLLMs, we recognize that other notable web agent frameworks also play an important role in the broader ecosystem. For example, SeeAct [4] leverages multimodal LLMs to reason over both rendered visuals and HTML structure, combining action planning with precise element grounding. WebVoyager [5] similar to browser-use, takes a different approach, using GPT‑4V and screenshot annotations (“Set‑of‑Mark” [3] hints) to navigate real websites end‑to‑end. OWL [6] (Optimized Workforce Learning) is a general agent framework that orchestrates multiple specialized agents in parallel, enabling flexible workflow automation and ranking among the top performers on benchmarks such as GAIA.
> Here are the results of the above-mentioned frameworks, and we will add this table to the revised version of our paper. For a fair comparison, the backbone MLLM for all frameworks is GPT-4o:
>
> Agent Framework | Pass@1(%)|
> |---------|-----------:|
> |Browser-Use|5.7%|
> |SeeAct [4]|7.0%|
> |WebVoyager [5]|9.0%|
> |OWL [6]|10.0%|
>
>
>
> [1] Koh JY, Lo R, Jang L, Duvvur V, Lim MC, Huang PY, Neubig G, Zhou S, Salakhutdinov R, Fried D. Visualwebarena: Evaluating multimodal agents on realistic visual web tasks. arXiv preprint arXiv:2401.13649. 2024.
>
> [2] Song Y, Ou T, Kong Y, Li Z, Neubig G, Yue X. VisualPuzzles: Decoupling Multimodal Reasoning Evaluation from Domain Knowledge. arXiv preprint arXiv:2504.10342. 2025.
>
> [3] Yang J, Zhang H, Li F, Zou X, Li C, Gao J. Set-of-mark prompting unleashes extraordinary visual grounding in gpt-4v, 2023.
>
>
> [4] Zheng B, Gou B, Kil J, Sun H, Su Y. Gpt-4v (ision) is a generalist web agent, if grounded. arXiv preprint arXiv:2401.01614. 2024.
>
> [5] He H, Yao W, Ma K, Yu W, Dai Y, Zhang H, Lan Z, Yu D. Webvoyager: Building an end-to-end web agent with large multimodal models. arXiv preprint arXiv:2401.13919. 2024.
>
> [6] Hu M, Zhou Y, Fan W, Nie Y, Xia B, Sun T, Ye Z, Jin Z, Li Y, Chen Q, Zhang Z. Owl: Optimized workforce learning for general multi-agent assistance in real-world task automation. arXiv preprint arXiv:2505.23885. 2025.

---

> ### Comment · Reviewer_Hkz5 · 2025-08-06
>
> So Tab. 4 are used for decomposing steps for human actions, not browser use steps?
>
> I am still confused about how it is estimated for browser use. I may regard browser use as a agent, that may itself decide to use tools from a predefined sets. In the rebuttal you only mention the "Set of markers" tool. Do you only divide by the use of this tool, or as in L116-118, any actions including interpreting visual content / planning a subgoal also forms a depth decomposition?
>
>
> If there are any misunderstanding about the method or any background, please help me correct it.

---

> > ### Author Response · Authors · 2025-08-06
> >
> > We appreciate the reviewer's follow-up question. To clarify, the decomposition framework in Table 4 is applied to both human actions and the browser use agent trajectories. The "Set-of-Mark" segmentation tool mentioned in the rebuttal is only one example, the agent has access to a wide range of tools, including *click_element_by_index*, *extract_content*, *drag_drop*, and others.
> >
> > Indeed, browser use is treated as an agent task, where an agent (e.g., a multimodal LLM) autonomously decides which tools to invoke based on its understanding of the current state. In a single action step, the agent chooses to invoke multiple tools as part of its reasoning and interaction process.
> >
> > Importantly, our definition of reasoning depth goes beyond just tool usage. It includes all semantically meaningful actions, such as interpreting visual content, planning subgoals, filtering options, and making visuomotor decisions. These are crucial components of the agent's cognitive trajectory and are counted in the step-wise decomposition.
> >
> > To make this concrete, we further provide the following example:
> >
> > If the agent performs the following in one turn:
> > * Action Step 1: Click next arrow to next image and get its src.
> >
> >      Tool 1: {"click_element_by_index": {"index": 1}}
> >
> >      Tool 2: {"extract_content": {"goal": "option image src", "should_strip_link_urls": false}}
> >
> > * Action Step 2: Click Submit button to validate match.
> >
> >      Tool 1: {"click_element_by_index":{"index":2}}
> >
> >      Tool 2: {"wait":{"seconds":1}}
> >
> >      Tool 3: {"extract_content":{"goal":"read updated stats","should_strip_link_urls":false}}
> >
> >
> > Though it totally **invokes three types of tools (five tool callings)** across these action steps, we count this as **two reasoning steps** because each reflects a distinct act of decision-making or goal progression. After the full interaction, we sum all distinct reasoning steps across turns to compute the final reasoning depth.
> >
> > We hope this clarifies how our decomposition captures both tool usage and the underlying reasoning process. Please feel free to let us know if you have any further questions and we are happy to address them.

---

> > > ### Comment · Reviewer_Hkz5 · 2025-08-09
> > >
> > > Thanks for your explanation. My questions are now cleared. I keep a positive attitude towards this paper.

---

### Comment · Area_Chair_gGDS · 2025-08-04
**Please read the author response and engage in discussion with the authors ASAP**

Dear Reviewers,

We are more than halfway through the author-reviewer discussion period. If you have not done so already, please carefully read the author response to your review, as well as the other reviews and the author responses to those too. If you have any follow-up questions for the authors, please post them ASAP so there is time for back and forth discussion with the authors.

If you don’t have any additional questions and if your concerns have been addressed, please update your ratings and the final justification accordingly. If you do not update your ratings, please explain in your final justification which concerns are still not addressed.

Note that the author-reviewer discussion period will end on August 6th, 11:59pm AoE.

Best,
Your AC

---

### Note · Authors · 2025-08-12

Dear Area Chair and Reviewers,

We sincerely thank you for your thoughtful comments, constructive suggestions, and engaging discussions during the review process. In our rebuttal, we have provided detailed clarifications to all questions and incorporated new experiments, analyses, and dataset extensions. All of these improvements will be integrated into the revised manuscript.

We are encouraged that the reviewers recognized the novelty and importance of our work. There is a clear consensus [Hkz5, AAv3, uzRf] that Open CaptchaWorld fills a critical gap by focusing on interactive CAPTCHAs, which is an underexplored but practical bottleneck for deploying web agents, and that our CAPTCHA Reasoning Depth metric offers a meaningful way to quantify multi-step cognitive and visuomotor challenges. Reviewers also appreciated our comprehensive evaluation, open-source release, and detailed failure analysis.

In response to specific feedback, we have:

1. Clarified the distinction between CAPTCHA challenges and other vision-agent benchmarks such as VisualWebArena, emphasizing the unique demands of non-literal reasoning, robustness against obfuscation, and fine-grained visuomotor control [Hkz5].

2. Expanded our explanation of Reasoning Depth with concrete examples, annotation guidelines, and step-decomposition methodology, addressing concerns about subjectivity [AAv3, Hkz5].

3. Compared multiple web agent frameworks (Browser-Use, SeeAct, WebVoyager, OWL) under a consistent backbone to isolate reasoning capability differences [AAv3, Hkz5].

4. Added new experimental results for Qwen2.5-VL-72B-Instruct and doubled our dataset size to enhance coverage and statistical robustness [uzRf].

We are grateful for the reviewers’ advice, which has helped refine our presentation and strengthen the contribution of Open CaptchaWorld. We hope the clarifications, new experiments, and dataset expansion have addressed all concerns satisfactorily.

Sincerely,
The Authors

---

### Decision · Program_Chairs · 2025-09-18

**Decision:**

Accept (poster)

**Comment:**

All reviewers recommend acceptance of the paper in their final ratings: borderline accept, accept, accept. The reviewers had raised some questions and concerns in their original reviews which the authors addressed in the rebuttal, leaving the reviewers convinced.

Upon carefully reading the reviews, the rebuttal and the reviewer-author discussion, the AC recommends accepting the paper as the paper proposes a novel and interesting task, a valuable benchmark covering wide range of captcha types, a reasoning depth metric, and presents a comprehensive evaluation of existing MLLMs. The AC thinks the paper has the promise to advance the state of art in web agents.

The AC recommends the authors to incorporate the feedback from the reviewers in the camera ready version.

===== FINAL UPDATE FROM DB Track PCs ====

The final decision for this paper has been taken by the program chairs after consultation with the SACs. All Senior Area Chairs have ranked papers according to the feedback from the AC during the review process. We decided to leave the original meta-review to reflect the opinion of the AC in light of the initial discussions with reviewers and SAC.